# Classification of Prefrontal Cortex Activity Based on Functional Near-Infrared Spectroscopy Data upon Olfactory Stimulation

**DOI:** 10.3390/brainsci11060701

**Published:** 2021-05-26

**Authors:** Cheng-Hsuan Chen, Kuo-Kai Shyu, Cheng-Kai Lu, Chi-Wen Jao, Po-Lei Lee

**Affiliations:** 1Department of Electrical Engineering, National Central University, No.300, Zhongda Rd., Zhongli District, Taoyuan City 32001, Taiwan; 048853@mail.fju.edu.tw (C.-H.C.); kkshyu@ee.ncu.edu.tw (K.-K.S.); 2Department of Electrical Engineering, Fu Jen Catholic University, No.510, Zhongzheng Rd., Xinzhuang District, New Taipei City 242062, Taiwan; 3Institute of Health & Analytics for Personalised Care, Universiti Teknologi PETRONAS, Seri Iskander 32610, Perak, Malaysia; chengkai.lu@utp.edu.my; 4Institute of Biophotonics, National Yang-Ming University, No.155, Sec. 2, Linong Street, Taipei 11221, Taiwan; c3665810@ms24.hinet.net

**Keywords:** functional near-infrared spectroscopy, olfaction, hemoglobin response function, support vector machine, classification, machine learning technique, prefrontal cortex

## Abstract

The sense of smell is one of the most important organs in humans, and olfactory imaging can detect signals in the anterior orbital frontal lobe. This study assessed olfactory stimuli using support vector machines (SVMs) with signals from functional near-infrared spectroscopy (fNIRS) data obtained from the prefrontal cortex. These data included odor stimuli and air state, which triggered the hemodynamic response function (HRF), determined from variations in oxyhemoglobin (oxyHb) and deoxyhemoglobin (deoxyHb) levels; photoplethysmography (PPG) of two wavelengths (raw optical red and near-infrared data); and the ratios of data from two optical datasets. We adopted three SVM kernel functions (i.e., linear, quadratic, and cubic) to analyze signals and compare their performance with the HRF and PPG signals. The results revealed that oxyHb yielded the most efficient single-signal data with a quadratic kernel function, and a combination of HRF and PPG signals yielded the most efficient multi-signal data with the cubic function. Our results revealed superior SVM analysis of HRFs for classifying odor and air status using fNIRS data during olfaction in humans. Furthermore, the olfactory stimulation can be accurately classified by using quadratic and cubic kernel functions in SVM, even for an individual participant data set.

## 1. Introduction

Support vector machines (SVMs) help classify human data from functional near-infrared spectroscopy (fNIRS) signals, and this method has novel applications [1,2], including a recently developed neuroimaging method for evaluating neural activity signals in brain cortical regions [3]. The classification accuracy of the eyes-open paradigm is 77.0%; that of the eyes-closed paradigm is 75.6% [4]. The performances and outcomes of two linear classifiers reportedly benefit from shrinkage linear discriminant analysis. fNIRS can be performed with the eyes open or closed during the activation of the prefrontal cortex (PFC) during a mental arithmetic task [5]. A three-choice system-paced single trial with separate mental arithmetic and mental signing tasks in the no-control state has been developed, and a linear classifier achieved overall classification accuracy of 56.2% [6]. Numerous studies have reported that an SVM algorithm using fNIRS signals may have low accuracy. Furthermore, studies on olfaction have extensively used SVMs or other algorithms [7,8,9,10]. However, few studies have used fNIRS signals to measure olfactory stimuli or the air state with an SVM algorithm.

fNIRS is used to investigate brain activity in healthy individuals, whose PFCs are stimulated through an olfactory stimulation task, and odorant intensity affects the hemodynamic response function (HRF) of oxyHb concentration [11,12]. Olfactory stimulation with an olfactometer using essential oils reportedly increases oxyHb levels and induces physiological and psychological relaxation [13]. Activation with strong fNIRS signals of the HRF in the PFC indicated olfactory stimulation when neonates were fed breast milk [14,15]. Using several odorants, Azuma et al. reported significant PFC activity during olfactory stimulation among patients with multiple chemical sensitivity (MCS) [16]. They reported olfactory stimulation by two odorants (sweet and fecal) at three concentrations (zero, odor recognition threshold, and normal perceived odor level) among MCS patients after fNIRS [17]. On identifying PFC activation due to olfactory stimulation, Azuma et al. reported that fNIRS is a valuable method for human testing [18].

Arterial blood oxygen saturation (SpO2) is a noninvasive dermal measurement, and photoplethysmography (PPG) signals of arterial pressure waves can be recorded at two wavelengths through fNIRS [19,20,21]. The pulse signal is measured using a light source with two wavelengths, red (PPG-R) and near-infrared (PPG-IR), and an optoelectronic detector detects the levels of light transmitted and reflected by human tissue under illumination [22].

The PPG waveform involves peak-to-peak amplitudes and has a pulsatile (AC) physiological waveform attributed to changes in the cardiac cycle from pumping a particular volume of blood with each baseline heartbeat and a gradually varying (DC) waveform along with various lower frequency components in venous blood, skin, and tissue [23,24]. The pulse signal ratio comprises two wavelengths (PPG-R and PPG-IR) of AC/DC and constitutes dbRatio data.

We used three SVM kernel functions (i.e., linear, quadratic, and cubic) to analyze and classify fNIRS signal data for olfactory activity. Signals were acquired from the PFC, where olfactory activity occurs. We acquired and transformed optical raw data to obtain a hemodynamic response (oxyHb and deoxyHb), PPG signal (PPG-R and PPG-IR), and dbRatio data, and then we used them to extract data features. The results of the preliminary trials using three kernel functions for SVM training are presented as performance accuracy and receiver operating characteristic (ROC) curves. The present SVM analysis of the HRF could serve as a novel method to classify odor and air status in human olfaction by using fNIRS.

## 2. Materials and Methods

### 2.1. Participants

Nine healthy individuals (seven male and two female; average age, 28 years) volunteered to participate in this study. The study design was explained to all participants. They had normal olfactory abilities and could smell lavender. Each was seated in a comfortable armchair. They heard white noise over earphones to reduce the effects of sound from the olfactometer; then, a bottle containing an odorant was opened.

### 2.2. System Architecture

The discussion of the system architecture (Figure 1a) is divided into three sections. The first section involves the acquisition and storage of data with an embedded NIRS device and a laptop computer, which monitor signals over the PFC in real time over two channels. The second section describes an experiment performed using lavender as an olfactory stimulus presented via an oxygen mask with an attached olfactometer and with an informative message relayed to the participants using Presentation software (Neurobehavioral Systems Inc., Berkeley, CA, USA). The third section describes the data analysis. Application of the raw data is discussed in two sections: One describes the incorporation of the raw data into MATLAB software to determine values estimating the percentage of oxygen bound to Hb; the other describes the analysis of the fNIRS signals in Homer2 to obtain estimates of brain activity.

### 2.3. fNIRS Analysis

We developed a flexible and simple NIRS protocol for monitoring brain activity. Our mainboard microcontroller, the STMicroelectronics Cortex-M4 Core Chip STM32F429, and daughter boards, AFE4490 models, were used for fNIRS data acquisition. The microcontroller functions at a 180 MHz processor clock frequency. The system had 21 communication interfaces including six serial peripheral interfaces (SPIs) and four universal asynchronous receiver/transmitters (UARTs), which could be configured through software through general-purpose input/output (GPIO). We used SPIs to read the analog-to-digital converter values and raw data transmitted by the microcontroller UART for storage on the laptop computer. The GPIO received triggers from the olfactometer during the experiment.

To minimize development cost and time, we used shelf components including customized printed circuit boards and two D-sub miniature connectors compatible with the Nellcor MAXN sensor. The light sensor has two sources, NIR and red light-emitting diodes (LEDs), that transmit light through the PFC. Unabsorbed light is received by a photodetector; the sample rate was 500 Hz in the channel. We used a Texas Instruments Low-Cost Pulse Oximeter with an AFE 4900 analog chip to detect brain activity. The AFE4490′s internal timer initiates the LED; the transceiver causes the light source to flash and detects the brain signal fed back.

### 2.4. Optode Settings

The LED emitter–detector pairs of two channels were positioned on the PFC to detect olfactory activity and rest signals, with two channels separated at the left and right side; the emitter-to-detector distance was 2.5 cm. The emitter had two wavelengths (λR and λNIR), 660 and 960 nm. The optode placement and channel location at Fp1 and Fp2 represent the reference points of the International 10–20 System (Figure 1b).

### 2.5. Stimulus Paradigm

The odor stimulus baseline was 35 s, olfactory stimulant time was 12 s, and rest time was 30 s; each trial comprised 16 blocks with a 20 s interval between blocks. Participants received a 17 s informative message initiated 5 s before the stimulus and terminated 12 s after, and they were given 25 s to answer the questions (Figure 2). These trials employed the oddball paradigm design, with a 12 s stimulus (opening bottles), which was a lavender odorant or distilled water; the stimuli were assigned at a 50:50 ratio. The stimulus block assignment followed the pattern 10101001, where 1 was lavender odor and 0 was water. The first prompt on the laptop computer screen read “Prepare to smell” for 17 s, and the second read “Did you smell?” after the stimulus. The participants in all blocks had 25 s to answer the question using a mouse. The participants were informed beforehand of the experimental paradigm about olfactory stimulant with air and odor and texts on computer screen but not about the regularity whole blocks.

### 2.6. Olfactory Delivery System

The olfactometer (ETT, Hershey, PA, USA) used for the lavender odorant was delivered as a fully automated system consisting of a set of Teflon tubing that transported odor and air, using an air pump, through an array of odorant chambers with solenoid valves to both nostrils of the participants. The timing and termination of odorant delivery were automated in accordance with the stimulus paradigm. During data acquisition, clean, moisturized compressed room air with or without an odorant was delivered through the tubing into an oxygen mask approximately 3 cm from the participant’s nose at a constant air flow rate of 8 L/min. The lavender intensity was set by diluting lavender oil to 2.0% intensity.

### 2.7. Signal Processing and Data Acquisition

Signal processing was performed in four stages (Figure 3): first, raw data were acquired; second, raw data were filtered, transforming the data into a hemodynamic response and PPG signals, and classifying the data into blocks; third, features were extracted from all data and the training data were labeled; fourth, classification and data statistics were performed using the SVM.

To determine the hemodynamic response from raw data, specified red and infrared wavelengths of light were transmitted to the scalp and then scattered through the cortex, where chromophores of oxyHb and deoxyHb, which absorb some NIR light, are present. Brain activity was detected by the detectors, raw intensity was recorded from the data transferred from the detectors, and the voltage was measured.

Signal processing filters were applied to the raw data. Thereafter, the data were transformed into hemodynamic responses; PPG signals and blocks of time were added to create block data in the second stage. The intensity data were then used to determine changes in the optical density (delta-optical density). The Homer2 software tool was used to analyze the hemodynamic responses of data associated with oxyHb and deoxyHb concentrations. The SpO2 codes could be used to determine PPG curves in two wavelengths, red and infrared (PPG-R, PPG-IR), and are derived from the dbRatio, as follows [25,26,27]:(1)$$\mathrm{dbRatio}=\frac{\;\mathrm{Red}\left(\mathrm{AC}/\mathrm{DC}\right)}{\mathrm{IR}\left(\mathrm{AC}/\mathrm{DC}\right)}$$
where AC and DC are the peak-to-peak amplitudes and baseline of the PPG pulse, respectively.

Data were recorded in 33 s blocks at a sample rate of 500 Hz; those data were the same as those recorded with a stimulus time of 3 s before and 30 s after. The locations of changes in the PPG-R, PPG-IR, dbRatio, oxyHb, and deoxyHb signals in each channel were observed in nine participants. The dbRatio signals were recorded for 33 s, as shown in Figure 4b.

### 2.8. Hemodynamic Response Function

Neuronal activation can be convolved with an HRF in the brain obtained through vascular imaging modalities based on stimuli and assessed through functional magnetic resonance imaging and fNIRS [28,29,30]. Numerous studies have attempted to use this hemodynamic response to better understand the mechanisms underlying neurovascular and neurometabolic coupling. The fNIRS signal intensity is altered during brain stimulation. OxyHb is detected through an increase in its concentration. The opposite is true of deoxyHb, which can be detected through a reduction in its concentration during brain activation. Increases in neural activity beyond the baseline level increase oxyHb and decrease deoxyHb levels [31,32], as show Figure 5 and Figure 6. In the present study, fNIRS was performed to assess the changes in the HRF of oxyHb and deoxyHb concentrations in accordance with the modified Beet–Lamberts law (BLL). The BLL algorithm in Homer2 software was used such that optical data represented the Hb concentration [33,34].

### 2.9. Feature Extraction

Feature data during signal processing were extracted during the third stage. We harnessed the following statistical properties of the five time-domain signals,

PPG-R, PPG-IR, dbRatio, oxyHb, and deoxyHb, as follows:Maximum (Max.);Minimum (Min.);Average (Mean.);Amount (Sum.);Trapezoidal numerical integration (Integ.).

These feature data have five signals and five features, separated by the two stimulus states of odor and air data, and status labels were assigned in a table for SVM training. Two types of data were used for training: a five-type signal yielding a single data point and a nine-type mixed signal yielding multiple data from a combination of two or three signals. The training results are listed as 14 types of data in Table 1, Table 2 and Table 3.

All the signal data were normalized before feature extraction using the following Z-Score formula [35].

### 2.10. Support Vector Machine

In the fourth stage of signal processing, the SVM was trained to classify data points. In particular, the SVM helped to identify the hyperplane, which provides a maximum margin between the plane surface and positive and negative points. This separating hyperplane was used as a frontier for classification, which becomes optimal at closed points, and is referred to as an SVM. The SVM was constructed by solving a dual optimization problem as follows:(2)$$maximize\;\overset{N}{\underset{i=1}{\sum }}a_{i}-\frac{1}{2}\overset{N}{\underset{i,j}{\sum }}a_{i}a_{j}y_{i}y_{j}K\left(x_{i}x_{j}\right)\;\mathrm{subject}\;\mathrm{to}\;\sum_{i=1}^{N}a_{i}y_{i}=0\;,\;0\leq a_{i}\leq C\;\forall \;i=1,\ldots N.$$

The coordinates (*x*_*i*,*y*_*i*) are training samples, and C is the penalty parameter for the slack variable that should be minimized [36]. The maximized equation of *K*(*x*_*i*,*x*_*j*) is the kernel function used to embed the training samples in an n-dimensional space. The kernel function affects the accuracy of the SVM classifier, which largely depends on the type used; several kernel functions are available for nonlinear mapping of input patterns. We used the polynomial kernel function to train and test our SVM classifier. Nonlinear transformation with the kernel function was performed as follows:(3)$$\mathrm{Polynomial}\;K\left(x_{i},x_{j}\right)={\left(1+x_{i}x_{j}\right)}^{\rho }$$
where *ρ* is the order of the polynomial.

The kernel function parameter ρ in Equation (4) and the margin-loss trade-off C in Equation (3) were considered to determine the optimal SVM performance and obtain an optimal C that the classifier can predict from unknown data or testing data. Optimal values of parameters were obtained through cross-validation using a grid search algorithm. The SVM was originally a binary classifier; therefore, we intended to separate the status of the two stimuli, odor and air [37,38].

### 2.11. Classification Evaluation

The accuracy, specificity, and sensitivity of classification performance were assessed using the true and false positive (TP, FP) rates and true and false negative (TN, FN) rates. Accordingly, the accuracy, specificity, and sensitivity were determined as follows [39]:(4)$$\mathrm{Accuracy}=\frac{\mathrm{TP}+\mathrm{TN}}{\mathrm{TP}+\mathrm{TN}+\mathrm{FP}+\mathrm{FN}}$$
(5)$$\mathrm{Sensitivity}=\frac{\mathrm{TP}}{\mathrm{TP}+\mathrm{FN}}$$
(6)$$\mathrm{Specificity}=\frac{\mathrm{TN}}{\mathrm{TN}+\mathrm{FP}}$$

Classification performance was analyzed on the basis of the hemodynamic response curve data, include PPG signals (PPG-R, PPG-IR, and dbRatio) and hemodynamic response signals (oxyHb and deoxyHb; Table 1, Table 2 and Table 3).

### 2.12. ROC Curve Analysis

ROC curves help determine the potential of various statistical methods. An ROC curve is a plot of a test’s TP rate (sensitivity) versus its FP (specificity). Thus, an ROC curve describes the trade-off between the sensitivity and specificity of the test, based on changes in the “negative” and “positive” results. The area under the ROC curve (AUC) is a quantitative measure of selectivity (e.g., 1.0 for the best selectivity; 0.5 for the worst selectivity) and relative accuracy values (ratio of TP and TN to all test samples). It is a function of both the sensitivity and specificity of a test, considering the entire range of errors. We performed ROC analysis of our SVM-based classification [40,41,42].

### 2.13. Data Analysis

Data were analyzed using the SVM algorithm and MATLAB 9.5 (The MathWorks Inc., Natick, MA, USA).

## 3. Results

The PFC hemodynamic responses of nine participants were analyzed. There were 9 subjects with a total of 25 experiments, and data were available from each of the 23 experiments, one from each experiment. Two of the subjects had two experiments each for a total of 4 times, and the remaining seven had three experiments each for a total of 21 times. We excluded two experiments from the seventh because the smell of lavender made the subjects doze off easily and caused no signal. The remaining 23 trials were included in the training dataset, and 16 blocks (eight odor and eight air blocks) were included at random in the trial. The training data included 368 block datasets, with 184 data for odor or air status. On analyzing the overall datasets, 16,500 data points were in each 33 s block (500 Hz).

Figure 7a displays the performance of a single signal of oxyHb on the ROC curve, for which the AUC of the quadratic and cubic kernel functions was 0.98 and 0.92, respectively. Figure 7b displays five signals of features in the training set for three kernel functions, where the performance of the ROC curve and SVM kernel function (quadratic and cubic) of the AUC were 0.72 and 0.65, respectively.

The ROC curve illustrates the performance of the classification model for optimal features, determined from the trade-off between sensitivity and specificity. The AUC quantifies the overall potential of the classifier to distinguish between stimuli. An ideal classifier has an AUC of 1, and a random classifier has an AUC of 0.5. Therefore, the larger the AUC, the better the performance of the classifier.

The specificity and sensitivity values were obtained by averaging the correspondence across one or more data sets. A comparison of results for single signals and multiple signals for three SVM kernel functions is depicted in Table 1, Table 2 and Table 3.

Figure 8 illustrates the single- and multi-signal accuracy of three kernel functions. Types 1 to 5 are single-signal data; Types 6 to 8 are two or three signals integrated into a table for the SVM training phase; Types 9 to 13 are multiple signals with four pieces of fNIRS data; Type 14 is integrated data from five signals. In detail, the types of signal data were as follows: (1) PPG-R; (2) PPG-IR; (3) dbRatio; (4) OxyHb; (5) deoxyHb; (6) PPG-R and PPG-IR; (7) HRF (oxyHb and deoxyHb); (8) PPG-R, PPG-IR, and dbRatio; (9) PPG signals and deoxyHb; (10) PPG signals and oxyHb; (11) HRF, PPG-R, and dbRatio (excluding IR); (12) HRF, PPG-IR, and dbRatio (excluding PPG-R); (13) HFR, PPG-R, and PPG-IR signals (without dbRatio); and (14) multiple signals with five fNIRS data and HRF.

As mentioned in the above tables, even in a smaller data set, the Quadratic or Cubic method revealed a superior performance in clustering than the linear method did (please refer to S7 in above table). By used linear method, the S7 of accuracy was only 50%, and the accuracy of clustering was prompt raised to 100% or 68.75% by using Quadratic or Cubic methods.

The best performance of clustering is S6 and the accuracy was 56.25%, 97.92% and 95.83 by using Linear Quadratic and Cubic methods, respectively in the Table 4. Furthermore, our results evidenced that the olfactory stimulation in prefrontal cortex activity can be accurately classified by quadratic and cubic kernel functions in SVM, even for an individual participant data set.

## 4. Discussion

We evaluated an innovative approach based on fNIRS signals for assessing olfactory signals in the PFC through SVM-based classification. Two HRF and PPG signals were evaluated using single or multiple signals from three SVM training kernel functions. The oxyHb of HRF displayed enhanced performance and accuracy. The fNIRS signal features were extracted for training, and these were primarily normal arithmetic features.

We used the simple Max., Min., Mean., Sum., and Integ. Feature extraction methods to accelerate the operation for embedded systems. We attempted to identify the most efficient set of features from participants’ brain activity [43,44]. Therefore, feature extraction from PPG and HRF signals was simplified through machine learning.

We used the Z-Score to normalize signals but disregarded the data processes that might influence accuracy. When the Z-Score was neglected (see Figure 9a), the oxyHb signals of accuracy with the cubic function dropped 84.38% to 53.13%, but the quadratic is not effected instead 87.5% up to 94.79%. The multi-signal (PPG and oxyHb) data resulted in accuracy of 70.83% which decreased to 59.38% with a quadratic function (Figure 9b). However, the Z-Score improved the accuracy results for two-signal data for the three kernel functions, as shown in Figure 9.

We have separated two kind of data for train set and test set. The total data was 368 available that 23 trials for each was 16 blocks. The training data were comprised of 272 blocks and the test data were comprised of 96 blocks; the cross-validation data which uses the “fitcsvm” function in MATLAB software by default used 10-fold the amount of training data. The test data set was taken from subject 1 to 6, each taking the second trial that the odor and air was eight blocks for every trial; the total was 96 blocks.

During the preparation of the training data, we used MATLAB software. Matrix transposition is an important step before feature extraction. Therefore, input data were arranged in a matrix of data acquired over time and recorded continuously in a column, and signal processing of the data in the column served as a stimulus block event. For example, the input data matrix dimensions were 16,500 × 8 (time points × blocks), which must be transposed to 8 × 16,500 (blocks × time points); otherwise, errors might be incurred during execution in MATLAB.

For sufficient training data, trial numbers from two to three trials were included for each participant; data from two participants who fell asleep during the experiment were excluded. In total, 727 s of trial data were acquired: 672 s constituted the toal time for 16 blocks (rest of 30 s and stimulus of 12 s), and 35 s of baseline and 20 s at the end were added. Keep awake, those participants to small, a question showed on laptop (Presentation^®^) and answered by mouse click. For related olfactory experiments, a shorter time may be preferable.

The fNIRS set of the optical (source–detector) pair at Fp1 and Fp2 (International 10–20 System) at the PFC is reasonable and facilitated the assessment of brain activity after the odor stimulus [11,12,13,14,15,16,17,18]. The olfactory complex comprises several areas, including the anterior olfactory nucleus, piriform cortex, olfactory tubercle, anterior cortical amygdaloid nucleus, periamygdaloid cortex, and entorhinal cortex, which receive inputs from the olfactory bulb and are present in a structurally distinct cortical region. The accessory olfactory bulb has medial and posterior cortical amygdaloid nuclei. Extrinsic axonal projections from the olfactory cortex to the orbital cortex, mediodorsal thalamic nucleus, hypothalamus, amygdala, and hippocampus are present [45]. The orbital cortex is a part of the PFC, brain area 13a of the posterior orbitofrontal cortex, and contains direct connections to the primary olfactory cortex unit [46].

Decreased olfactory function is common among patients with idiopathic Parkinson’s disease [47] or Alzheimer’s disease [48] and older individuals [49]. Olfactory function in humans plays important protective roles and contributes to interpersonal interactions, eating habits, and nutrient intake, which are severely altered in numerous patients with olfactory disorders [50]. Loss of olfactory function is present in approximately 50% of older individuals (65–80 years) and approximately 75% of those aged ≥ 80 years [51]. An early diagnosis of olfactory dysfunction through fNIRS is thus recommended.

In future studies, we intend to investigate other features extracted from HRF and PPG signals. Furthermore, fNIRS may be using embedded in a mobile device with a data center which big data processing online. Further studies are required that predict olfactory dysfunction among participants with SVMs or other machine learning techniques with large clinical datasets.

## 5. Conclusions

This study reports the potential of using an SVM for training five types of single fNIRS signal data and nine types of multiple fNIRS signal data, extracting features regarding olfactory stimuli, yielding the original HRF and PPG signals from fNIRS raw data. We developed an fNIRS system that measures raw data at the PFC through olfactory stimulation.

Among three SVM kernels, the quadratic function was optimal for fNIRS signals, with average accuracy of 63.61%. The cubicand linear kernel functions had average accuracy of 60.34% and 44.12%, respectively. Therefore, we used the quadratic kernel function.

For 14 types of data, including five pieces of single-signal data using the quadratic kernel function, the accuracy of oxyHb was the highest at 87.5%, whereas the accuracy for PPG-R was 47.92%, lower than that of the others. Multiple oxyHb and PPG signals (PPG-R, PPG-IR, and dbRatio) with the quadratic kernel function resulted in an adequate accuracy of >70.83%. Type 11–14 resulted in an accuracy of >67%; however, Types 6 and 8 included PPG but excluded oxyHb, and the accuracy decreased to 55.21% and 57.29%. Type 5, deoxyHb, resulted in the lowest accuracy, at 48.96%. The oxyHb signal affected the performance accuracy of both the single- and multi-signal data for all three SVM kernels.

In summary, fNIRS signals were successfully assessed to evaluate olfactory stimulation, and simple extracted features were employed for SVM training with three kernel functions to assess valuable data.

## Figures and Tables

**Figure 1 brainsci-11-00701-f001:**
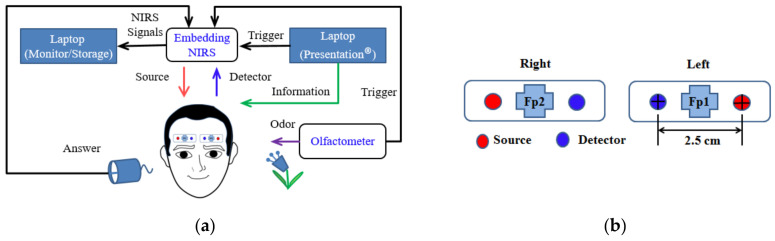
(**a**) Schematic of the study’s fNIRS system design. (**b**) Optode placement and channel location on the prefrontal cortex at reference points Fp1 and Fp2, in accordance with the International 10–20 System of placement. This system comprises two sources and two detectors, and each source–detector pair is separated by 2.5 cm.

**Figure 2 brainsci-11-00701-f002:**
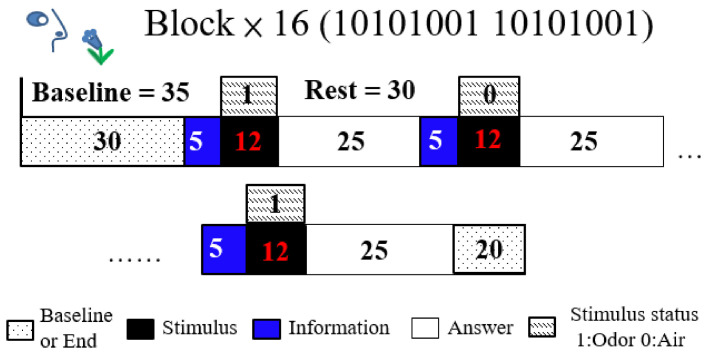
Schematic representation of the experimental paradigm: A block comprising a 12 s stimulus and a 30 s rest period, along with a 35 s baseline and 20 s termination period.

**Figure 3 brainsci-11-00701-f003:**
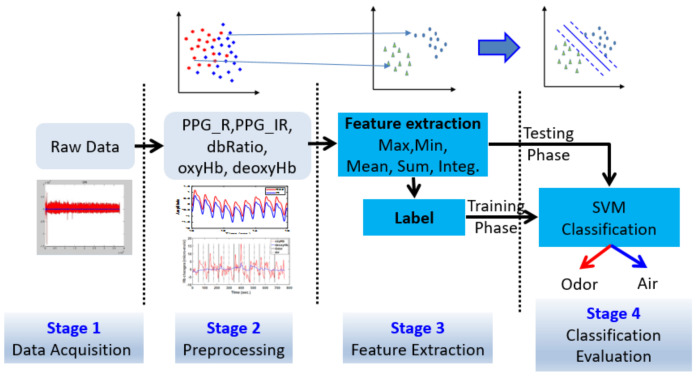
Flow chart of signal processing using the present machine learning approach to separate odor and air classes through photoplethysmography (PPG-R, PPG-IR, and dbRatio) and hemoglobin response function (including oxyHb and deoxyHb) signal data.

**Figure 4 brainsci-11-00701-f004:**
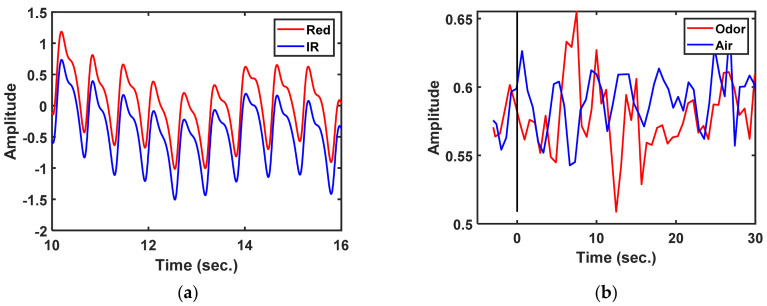
Functional near-infrared spectroscopy measurement of the prefrontal cortex: (**a**) raw photoplethysmography (PPG) PPG-R and PPG-IR signals, (**b**) 33 s block data in which the dbRatio waveforms have an odor and air status.

**Figure 5 brainsci-11-00701-f005:**
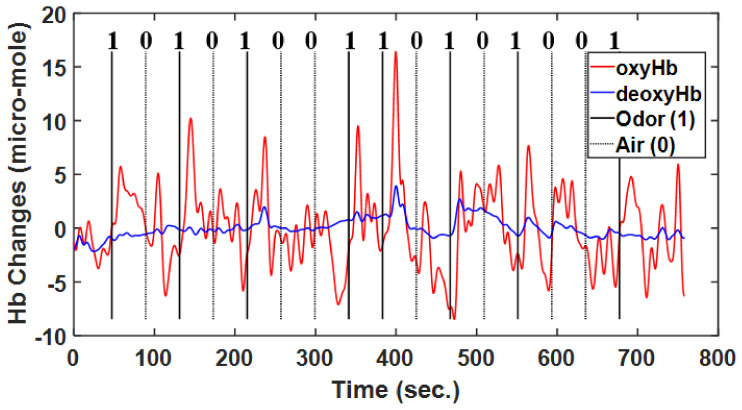
Typical example of raw hemodynamic response function including oxyHb and deoxyHb data, with air and odor as the stimuli, in accordance with the experimental paradigm.

**Figure 6 brainsci-11-00701-f006:**
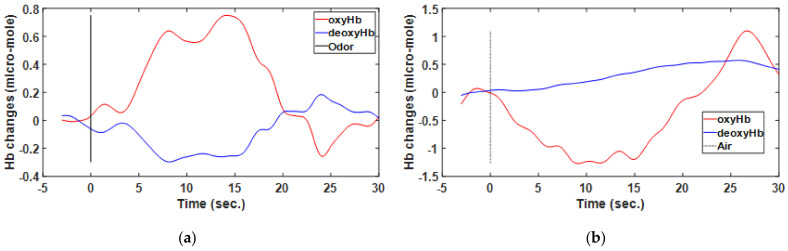
Hemodynamic response of one participant determined through functional near-infrared spectrometry. Subplots show the averaged results for the region of interest from among eight blocks wherein odor (**a**) and air (**b**) were used as individual stimuli.

**Figure 7 brainsci-11-00701-f007:**
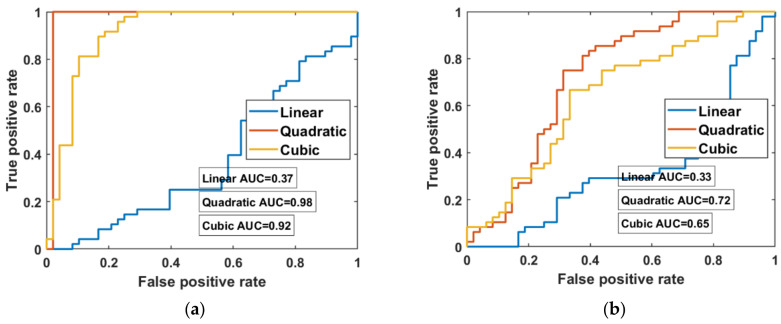
Receiver operating characteristic curves of signals of a dataset with performance of three kernels for an SVM classifier: (**a**) single signal of oxyHb using the quadratic kernel with an area under the curve of 0.998, and (**b**) combination of five signals with the optimal area under the curve of 0.72 with the cubic function.

**Figure 8 brainsci-11-00701-f008:**
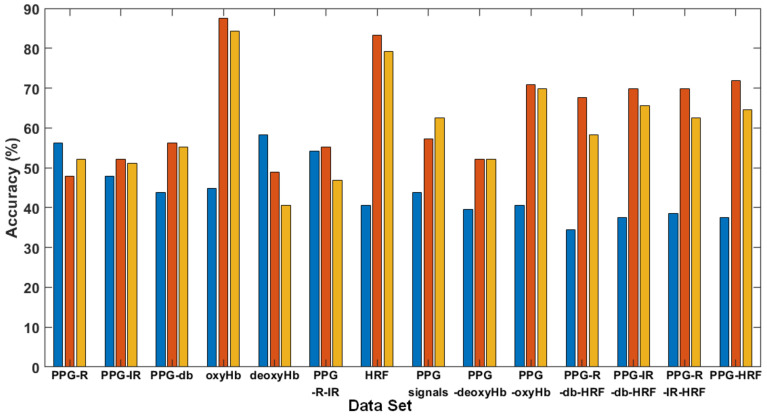
Accuracy comparison for 14 types of data and three kernel functions: linear, quadratic, and cubic.

**Figure 9 brainsci-11-00701-f009:**
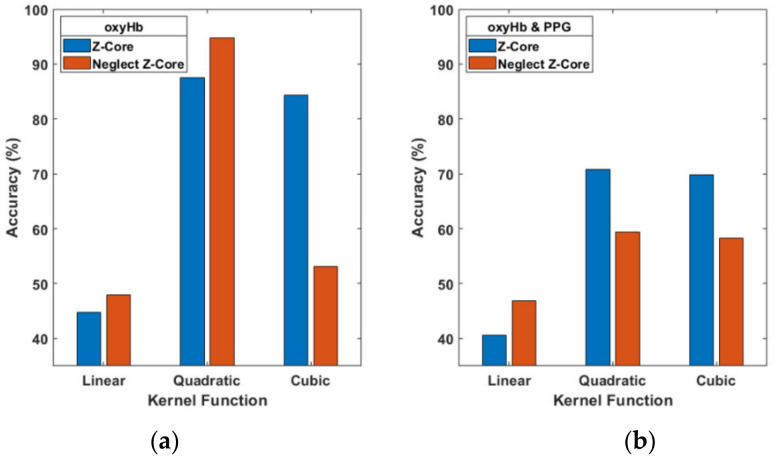
Comparison of the Z-Score and neglect Z-Score for single (**a**) and multiple signals (**b**) and three kernel functions.

**Table 1 brainsci-11-00701-t001:** Support vector machines with linear kernel function; all signals were photoplethysmography signals (PPG-R, PPG-IR, and PPG-dbRatio) and hemodynamic response function (oxyHb and deoxyHb) signals. The superscript asterisk is the best in table of accuracy.

Type of Data	Accuracy (%)	Sensitivity (%)	Specificity (%)	AUC
PPG-R	56.25	70.83	58.3	0.59
PPG-IR	47.92	72.92	33.33	0.47
PPG-dbRatio	43.75	4.16	100	0.42
oxyHb	44.79	0	100	0.37
deoxyHb	58.33 *	91.67	39.58	0.64
PPG-R and PPG-IR	54.17	77.08	47.92	0.59
HRF (oxyHb and deoxyHb)	40.63	97.92	4.17	0.32
PPG signals	43.75	91.67	12.5	0.4
deoxyHb and PPG signals	39.58	93.75	10.42	0.39
oxyHb and PPG signals	40.63	91.67	10.42	0.34
HRF and PPG (-R-dbRatio)	34.38	91.67	14.58	0.33
HRF and PPG (-IR-dbRatio)	37.5	100	2.08	0.32
HRF and PPG (-R-IR)	38.54	97.92	10.42	0.32
HRF and PPG signals	37.5	97.92	4.17	0.33

**Table 2 brainsci-11-00701-t002:** Support vector machines with quadratic kernel function; all signals were photoplethysmography signals (PPG-R, PPG-IR, and PPG-dbRatio) and hemodynamic response function signals (oxyHb and deoxyHb). The superscript asterisk is the best accuracy in table.

Type of Data	Accuracy (%)	Sensitivity (%)	Specificity (%)	AUC
PPG-R	47.92	66.67	52.08	0.56
PPG-IR	52.08	22.92	85.42	0.5
PPG-dbRatio	56.25	29.17	89.58	0.59
oxyHb	87.5 *	100	97.92	0.98
deoxyHb	48.96	60.42	54.17	0.53
PPG-R and PPG-IR	55.21	52.08	62.5	0.54
HRF (oxyHb and deoxyHb)	83.33	95.83	79.17	0.92
PPG signals	57.29	54.17	64.58	0.57
deoxyHb and PPG signals	52.08	72.92	50	0.55
oxyHb and PPG signals	70.83	79.17	64.58	0.77
HRF and PPG (-R-dbRatio)	67.71	75	66.67	0.69
HRF and PPG (-IR-dbRatio)	69.79	91.67	56.25	0.74
HRF and PPG (-R-IR)	69.79	77.08	68.75	0.75
HRF and PPG signals	71.88	75	68.75	0.72

**Table 3 brainsci-11-00701-t003:** Support vector machines with cubic kernel function; all signals were photoplethysmography signals (PPG-R, PPG-IR, and PPG-dbRatio) and hemodynamic response function signals (oxyHb and deoxyHb). The superscript asterisk is the best accuracy in table.

Type of Data	Accuracy (%)	Sensitivity (%)	Specificity (%)	AUC
PPG-R	52.08	20.83	85.42	0.5
PPG-IR	51.04	62.5	52.08	0.54
PPG-dbRatio	55.21	83.33	45.83	0.62
oxyHb	84.38 *	89.58	83.33	0.92
deoxyHb	40.63	0	100	0.38
PPG-R and PPG-IR	46.88	93.75	12.5	0.48
HRF (oxyHb and deoxyHb)	79.17	91.67	66.67	0.76
PPG signals	62.5	58.33	66.67	0.59
deoxyHb and PPG signals	52.08	68.75	47.92	0.57
oxyHb and PPG signals	69.79	72.92	72.92	0.73
HRF and PPG (-R-dbRatio)	58.33	75	52.08	0.65
HRF and PPG (-IR-dbRatio)	65.63	68.75	64.58	0.65
HRF and PPG (-R-IR)	62.5	89.58	43.75	0.67
HRF and PPG signals	64.58	66.67	66.67	0.65

**Table 4 brainsci-11-00701-t004:** Accuracies for individual participants with three kernel functions.

Participants	Linear Accuracy (%)	Quadratic Accuracy (%)	Cubic Accuracy (%)	Block Number	Train-Test Ratio
S1	65.63	90.63	84.38	32	91.3:8.7
S2	53.13	96.88	93.75	48	87:13
S3	56.25	90.63	90.63	32	91.3:8.7
S4	52.08	83.33	87.5	48	87:13
S5	48.44	90.63	76.56	48	87:13
S6	56.25	97.92	95.83	48	87:13
S7	50	100	68.75	16	95.7:4.3
S8	50	75	77.08	48	87:13
S9	54.17	89.58	81.25	48	87:13
Average	53.99	90.51	83.97	40.9	89:11

## Data Availability

The data are not publicly available due to the privacy concern raised by our IRB.

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
