# Peer review of "Classification of Prefrontal Cortex Activity Based on Functional Near-Infrared Spectroscopy Data upon Olfactory Stimulation"

_brainsci, 2021, doi:10.3390/brainsci11060701_

Round 1
Reviewer 1 Report
Thank you for the opportunity to read your manuscript. You have presented some interesting research where you classify fNIRS data to detect the presence/absence of a strong odor. This research has the potential to be useful beyond just the olfactory areas, and the methods could be adapted for other fNIRS measurements. However, the experiment is not presented well at present. The abstract/intro/body shows WHAT was done, but it is less clear WHY it was done. Presenting a stronger case would strengthen the work. i.e. it is not clear why we would want to classify odor and air status using fNIRS. Re-framing the work to show how it satisfies a need, and whether this technique could be useful for other aspects of fNIRS would help to build a justification for the work.
It is not clear whether the main novelty is in the fact that odor was classified, or that you found an optimal SVM algorithm, or that OxyHb is the best predictor. This would be best presented as part of the “story” that is currently missing from the paper.
The SVM appears to have been run on pooled data. It would have been interesting to see if there were higher accuracies for individual participants (or at least have this noted in the results).
There were a few areas where typos/grammatical oddities made reading very hard. E.g.
260-262 “Those participants all trials were 25, top second participants were four trials that per, other seven participants were 21. We excluded two trials in which no signals so signals were obtained because the lavender odor made the participants drowsy.” – need editing as incomprehensible and incorrect
338 – 339 “The focus of the experiment was for participants to lavender and answer a question by mouse click, without falling asleep.”
360 – 361 “Furthermore, fNIRS may be useful embedded in a mobile device with a data center for big data processing online.”
Figure 6: needs to be made clearer (title?) that one plot is air and one odor
Reviewer 2 Report
This article described a machine learning based method for detecting cortical activation in fNIRS studies. The topic and the proposed approach are of interest in general, however, there are some areas that need to be improved:
- The article does not provide sufficient information regarding the train/test/cross validation data breakdown. This is a very important detail for interpreting the presented results and understanding potential limitations. From figure 3 it seems like the same data is fed to the classifier both for test and training, and the text doesn’t provide information to clarify this. Also the performance metrics provided are a bit concerning which would raise question around training/test breakdown. In the abstract, the accuracy is reported as 99.46% with sensitivity of 100% and specificity of 98.91. I would like to make sure these numbers are from test dataset (data that has not been presented to the model before).
- The article in general could use an editorial proof reading to make it a little more clear (example lines 204, 260, 262 etc.)
- Reading through the abstract it is very hard to understand what exactly the article is proposing. It seems to me that the article is proposing an SVM based technique for detecting brain activation from fNIRS signal. This is not clear from the abstract.
- What is the motivation for using both Average and sum for each block as features? These two seem highly correlated.
- What is the motivation for using PPG in this study? Even though PPG uses similar principles as NIRS, the nature of the data it captures is different. Why do authors think the source of the PPG signal is from cortical tissue and not the superficial scalp tissue?
- What is the significance of the proposed method? For example assuming this method works well, what benefit does it provide compared to the standard general linear model (GLM)? It would be good to add a section about this in the discussion.
- Explanation of basics do not seem appropriate in a journal article (eg. paragraph starting at line 328)
Reviewer 3 Report
The paper discusses a classification of prefrontal cortex activity based on functional near-infrared spectroscopy data upon olfactory stimulation. The research approach is god and the explanation of hardware and software implementations are also good. However, the explanations of row data and the classifications can be improved by explaining why SVM was chosen for the classification job and how SVM can out-perform other methods.
Figure 8 appears in two places (lines 285 & 297) and correct that please.
Author Response
Please check attached file.

This manuscript is a resubmission of an earlier submission. The following is a list of the peer review reports and author responses from that submission.
Round 1
Reviewer 1 Report
Major comments:
The authors employed a custom-developed fNIRS system to investigate the hemodynamic activation in response to olfactory stimulation in 9 healthy participants out of which two dropped out. Authors employed SVM classification to classify two classes corresponding to odor and air. The paradigm employed is novel but the instrumentation, signal analysis and the representation of the results does not support the conclusions of the authors. Also, with such a smaller number of trials, the conclusion is too far-reaching.
The major limitations of this article are:
Very low sample size. Although the study was performed on 9 healthy participants two participants dropped out of the study. From the remaining 7 participants only 5 of them performed 3 trials. Therefore, the number of trials is too less. Custom NIRS device whose signal to noise ratio, sensitivity and specificity are unknown. To know the quality of the signal acquired it is very important to show the HRF from each participant. The hemodynamic response presented in Figure 6 does not seem like an HRF function.
Authors should show the averaged and single-trial HRF function which will let the reader appreciate the NIRS signal. Since we do not know the quality of HRF signal nothing much can be concluded from the classification result.
Minor comments:
The introduction seems to be redundant and needs to be rewritten. Please check the sentence construct throughout the article, many a time it is difficult to read and comprehend the sentences.
Reviewer 2 Report
I did notice that some english language has been improved but for the most part it is still very difficult to read the paper. Also I read the following sentence "Publisher’s note: Springer Nature remains neutral with regard to jurisdictional claims in 473 published maps and institutional affiliations". This does not make sense why Springer nature in involved in this publication. I think this paper still needs extensive editing before it can be accepted for publication here.